# Hemodynamic Characteristics of Cardiovascular System in Simulated Zero and Partial Gravities Based on CFD Modeling and Simulation

**DOI:** 10.3390/life13020407

**Published:** 2023-02-01

**Authors:** Lei Sun, Lijie Ding, Lei Li, Ningning Yin, Nianen Yang, Yi Zhang, Xiaodong Xing, Zhiyong Zhang, Chen Dong

**Affiliations:** 1Laboratory of Virtual Reality and Simulation Technology, Shandong Sport University, Jinan 250102, China; 2China National Football Academy, Shandong Sport University, Rizhao 276827, China; 3Department of Health Service and Management, Shandong Sport University, Jinan 250102, China; 4Department of BioMedical Engineering, University of Groningen, 9713 Groningen, The Netherlands; 5School of Sports Art, Shandong Sport University, Jinan 250102, China; 6College of Sports and Health, Shandong Sport University, Rizhao 276827, China

**Keywords:** partial gravities, hemodynamics, cardiovascular risk, random walk, CFD simulation

## Abstract

Zero and partial gravities (ZPG) increase cardiovascular risk, while the corresponding theoretical foundation remains uncertain. In the article, the ZPG were generated through a rotating frame with two degrees of freedom in combination with the random walk algorithm. A precise 3D geometric configuration of the cardiovascular system was developed, and the *Navier-Stokes* laminar flow and solid mechanics were used as governing equations for blood flow and the surrounding tissue in the cardiovascular system. The ZPG were designed into governing equations through the volume force term. The computational fluid dynamics’ (CFD) simulations in combination with proper boundary conditions were carried out to investigate the influences of ZPG on the distribution of blood flow velocity, pressure, and shear stress in the cardiovascular system. The findings show that as simulated gravity gradually decreases from 0.7 g to 0.5 g to 0.3 g to 0 g, as opposed to normal gravity of 1 g, the maximum values of blood flow velocity, pressure, and shear stress on the walls of the aorta and its ramification significantly increase, which would lead to cardiovascular diseases. The research will lay a theoretical foundation for the comprehension of the ZPG effect on cardiovascular risk and the development of effective prevention and control measures under the circumstance of ZPG.

## 1. Introduction

Zero and partial gravities (ZPG) correlate with an increased risk for cardiovascular diseases, which have been currently supported by much research. Some reports have demonstrated that there is a positive correlation between ZPG and cardiovascular risks. For example, some older people suffered sudden myocardial and cerebral infarction after going on a rollercoaster, a pendulum ride and a helter-skelter in amusement parks [1,2,3,4].

The physiological effects of microgravity on astronauts have been extensively researched in the field of aerospace medicine, and exposure to microgravity has been linked to a number of physiological alterations in astronauts. For instance, it was thought that microgravity played a significant role in increasing osteoclasts’ rate of bone resorption and decreasing the cellular integrity of osteoblasts [5]. Some astronauts living and working for a long time in the space station increased the risk of cardiovascular health problems after they got back to the ground, which was thought to be caused by the blood flow and pressure increase under a weightless condition, resulting in the chronic impairment of the cardiovascular system [6,7,8,9,10,11].

Current technology allows for the production and implementation of the ZPG effect, which has been extensively employed to study the physiological and behavioral reactions of higher plants, microbes, and animal tissues and organs to the ZPG effect [12,13,14,15,16,17,18]. For instance, a random positioning device that can continuously alter its orientation randomly with respect to an experiment’s gravity vector and produce outcomes similar to those of real microgravity when the direction changes occur more quickly than the body’s reaction time to gravity [13,19]. Rotating devices were used to simulate a microgravity environment, and it was discovered that microgravity caused human microvascular endothelial cells to apoptose, which was linked to the body’s reaction by downregulating the PI3K/Akt pathway, upregulating the expression of NF-κB, and depolymerizing F-actin [20].

Similarly, CFD modeling and simulation is extensively used to investigate physiological and pathological behavior of human’s cardiovascular system in microgravity. Through CFD simulations, it has been discovered that the microgravity causes an increase in stroke volume, upper body capillary pressure, and volume [21]. A low wall shear stress gradient was seen in left bifurcations with wide angles, which was based on CFD modeling and simulations, and revealed the direct correlation between coronary angulations and subsequent hemodynamic alterations [22].

As is known, the geometric configuration, governing equations and boundary conditions are the three elements of CFD modeling. Based on governing equations of *NS* laminar or turbulent flow, most of research focuses on how to change the geometric configuration of the cardiovascular system and boundary conditions resulting in the variation of hemodynamic characteristics under the normal and pathological state of cardiovascular system, without sufficiently taking the gravity term in *NS* governing equations into consideration.

Although numerous studies have been carried out to look into the relationship between ZPG and cardiovascular risk, the theoretic mechanism is still unknown at this time. In this article, the hypothesis has been proposed based on related documents [1,4,11,21] to address this problem as follows.

ZPG could significantly change hemodynamic characteristics, such as the elevation of blood pressure, flow acceleration, increase of shear stress on the vascular wall in the cardiovascular system, greatly promoting the risk of cardiovascular diseases. Traditionally, prototype experiments need to be carried out to confirm a proposed hypothesis. The cardiovascular system is recognized to be very complex, having nonlinear and time-variant characteristics that inevitably result in complex dynamic behaviors. Therefore, it is practically impossible to clearly elucidate the constitutive relationship between ZPG effects and cardiovascular risks through actual prototype experiments. Furthermore, prototype experiments involving human beings and animals often give rise to the ethical issues. Therefore, in the given situation, using computer simulation has been the most effective and cheap technique to address these complicated difficulties and evaluate the offered hypothesis [23]. According to the growing consensus in complex systems science, this may be the only method for methodically examining the complex structure and dynamic behaviors of biological systems [24].

Therefore, in the article, the modeling and simulation of computational fluid dynamics (CFD) was conducted to confirm the hypothesis proposed. The ZPG were realized through the rotating frame relative to the world frame with two degrees of freedom in combination with the random walk algorithm. Then the precise 3D geometric configuration of the cardiovascular system was developed. Governing equations including the equations of *Navier-Stokes* (*NS*) laminar flow and solid mechanics were selected and applied for blood flow and the surrounding tissue, and the simulated ZPG were coupled with governing equations through the volume force term. The distribution of blood flow velocity, pressure, and shear stress in the cardiovascular system was examined using CFD simulations once the necessary boundary conditions were set. The findings demonstrate that when the simulated ZPG gradually decrease in comparison with 1 g of normal gravity, the maximum values of blood flow velocity, pressure, and shear stress on the walls of the aorta and its ramification dramatically rise.

To our knowledge, no CFD model in use has, however, properly accounted for the various gravity gradients. Therefore, the goal of this study is to create a cardiovascular system CFD model that is highly valid and coupled with the simulated ZPG effect. The aorta’s hemodynamic properties and its ramifications under simulated ZPG have received particular attention.

## 2. Materials and Methods

### 2.1. Simulated Zero and Partial Gravities

The simulated ZPG is realized through a rotating frame relative to the world frame with two degrees of freedom (Figure 1).

The gravity direction rotates continuously with respect to the experimental frame since it is fixed in the world frame. A mean gravity that ranges from the hypothetical 0 g to 1 g can be calculated by averaging the gravity vector acceleration across time. Simulated ZPG are defined as a mean gravity of between 0 and 1 g. The set of angles (*α* and *β*) that make up the mechanism’s joint space create a two-plane. The task space is a sphere-shaped collection of gravity directions in relation to the experiment. The three parts of the gravity vector [*g_x_*, *g_y_*, *g_z_*] can be used to parametrize the sphere, which is embedded in three-dimensional space.

At the beginning of the rotation, the acceleration of gravity on the ground is 1 g with just sticking straight down, hence the gravity vector is expressed as a vector [0, 0, −1]; after a period of time *t*, the rotating frame rotated *α*(*t*) and *β*(*t*) on *x-* and *y*-axes, respectively. Based on the coordinate transformations, the gravity vector at time *t* could be calculated as follows:(1)$$\vec{g}\left(t\right)=\;\left(\begin{array}{c} g_{x}\\ g_{y}\\ g_{z} \end{array}\right)=\left(\begin{array}{ccc} \cos\beta \left(t\right) & 0 & \sin\beta \left(t\right)\\ 0 & 1 & 0\\ -\sin\beta \left(t\right) & 0 & \cos\beta \left(t\right) \end{array}\right)\cdot \left(\begin{array}{ccc} 1 & 0 & 0\\ 0 & \cos\alpha \left(t\right) & -\sin\alpha \left(t\right)\\ 0 & \sin\alpha \left(t\right) & \cos\alpha \left(t\right) \end{array}\right)\cdot \left(\begin{array}{c} 0\\ 0\\ -1 \end{array}\right)$$
where the first and the second term is the rotation matrix of *x*- and *y*- axes, respectively.

Obtain,
(2)$$\vec{g}\left(t\right)=\left(\begin{array}{c} -\cos\alpha \left(t\right)\cdot \sin\beta \left(t\right)\\ \sin\alpha \left(t\right)\\ -\cos\alpha \left(t\right)\cdot \cos\beta \left(t\right) \end{array}\right)$$

Set *α* = *ω_x_ t*, *β* = *ω_y_ t*, get
(3)$$\vec{g}\left(t\right)=\left(\begin{array}{c} -\cos\omega_{x}t\cdot \sin\omega_{y}t\\ \sin\omega_{x}t\\ -\cos\omega_{x}t\cdot \cos\omega_{y}t \end{array}\right)$$
where *ω_x_* and *ω_y_* is the angular velocity of *x*- and *y*-axes, respectively. The Equation (3) is the constitutive relation between gravity vector, $\vec{g}\left(t\right)$, and the rotation speed of axes. The $\vec{g}\left(t\right)$ can be averaged over time to obtain a certain mean gravity:(4)$$\mu \left(\vec{g}\left(t\right)\right)=\frac{1}{t}\int \vec{g}\left(t\right)dt$$
where *μ* = 0 is the necessary condition for the simulated ZPG. This is because a sphere is created by the space created by all gravity directions. This sphere is traversed by the gravity route, which stops at various locations. There is a particular distribution around the sphere formed by the relative times it spends in the areas around various spots. The gravity path $\vec{g}\left(t\right)$ converges to a uniform distribution if it visits every point equally, and to a non-uniform distribution if it visits certain sites more frequently than others. The fact that different distributions might result in the same mean gravity is important to note.

In order to measure for this uniformity, a specific standard deviation (*σ*) of the density of the distribution over spherical surface is defined as follows:(5)$$\sigma =\sqrt{\frac{1}{4\pi t}\int_{0}^{2\pi }\int_{0}^{2\pi }{\left(d\left(u,v\right)-\frac{t}{4\pi }\right)}^{2}cos\left(u\right)dudv}$$
where *u* and *v* are skew lattice coordinates on the surface of the sphere, and density *d*(*u*, *v*) is defined as the amount of time the gravity direction has spent in a neighborhood point (*u*, *v*) on the surface of the sphere. So, by controlling the combinations of *μ* and *σ* and produced by the random walks’ algorithm in this study, simulated ZPG may be obtained.

### 2.2. CFD Modeling and Simulation of Cardiovascular System under Simulated Zero and Partial Gravities

In the article, a section of the cardiovascular system was chosen for CFD modeling and simulation; in particular, the upper region of the aorta. The cardiac muscle contains the aorta and the blood arteries that branch from it. Blood flow puts pressure on the inside surfaces of the artery and its branches, causing the tissue to deform [25]. Because the cardiovascular system is complicated, the CFD modeling and calculations were performed according to the following assumptions for a simplified calculation:(1)The flow in the blood artery was considered to be a laminar flow, assuming that the cardiovascular system was in good health.(2)A mechanical analysis of the deformation of the tissue and artery, where it is assumed that any change in the geometric configuration is unaffected by the shape of the vessel walls and that the blood flow domain of the cardiovascular system is invariable with only a one-way weak fluid-structural coupling.

Based on assumptions, the research was carried out by following steps.

Firstly, the accurate geometric configuration of the cardiovascular system was drawn by *SolidWorks*, and then imported to *Comsol Multiphysics* for CFD modeling and simulation.

Secondly, the 3D domain and surface of the geometric configuration of the cardiovascular system were used to specify the governing equations and boundary conditions, respectively. Two governing equations were applied for fluid-structural coupling based on the mechanisms of cardiac hemodynamics to examine the effects of simulated ZPG on the distribution of blood flow velocity, pressure, and shear stress on the cardiovascular system. The gravity vector (Equation (3)) is tightly coupled by the term of volume force of the governing equations.

(1)Laminar flow

The classic *Navier-Stokes* equations are solved in the blood domain as follows:(6)$$\rho \frac{\partial u_{f}}{\partial t}+\rho \left(u_{f}\cdot \nabla \right)u_{f}=\nabla \cdot \left[-pl+\mu \left(\nabla u_{f}+{\left(\nabla u_{f}\right)}^{T}\right)\right]+F$$
(7)$$\rho \nabla \cdot \left(u_{f}\right)=0$$
where the dependent variables, *u_f_* (m s^−1^) and *p* (Pa), stand for the blood flow velocity and blood vessel pressure, respectively. Blood’s density and dynamic viscosity were symbolized, respectively, by the letters *ρ* and *μ*.

The volume force term F, i.e., $F=\vec{g}\left(t\right)$ were used to account for the effects of the simulated ZPG on the distribution of blood flow velocity, pressure, and shear stress on the cardiovascular system. According to Equation (7), the blood flow is incompressible.

(2)Solid mechanics

Mechanical analysis must be taken into account since the heart muscle exhibits a stiffness that prevents arterial deformation brought on by the applied pressure. As a result of computing the overall stress distribution during the fluid dynamics study, the model expresses the load as follows:(8)$$\rho_{s}\frac{\partial^{2}u_{s}}{\partial t^{2}}=\nabla \cdot S+\mathrm{Fs}$$
where *ρ_s_* is the material density, S is the strain, Fs is the volume force with components in the actual configuration (spatial frame) given with respect to the deformed volume. Similarly, $F_{s}=\vec{g}\left(t\right)$.

According to the real physiological state of the rotating cardiovascular system, boundary conditions, such as the Dirichlet and Neumann conditions, were properly stated [26].

Finally, using the platforms of Matlab [27], Comsol Multiphysics [28], and their interface kit (Comsol Multiphysics with Matlab), digital simulations were carried out to examine blood flow, pressure, and shear stress characteristics in the domain and wall of the blood vessel under various simulated partial gravities realized by proper rotation of the cardiovascular system.

## 3. Results and Discussion

### 3.1. Simulated ZPG Realized by Random Walk

The proposed gravity path is a random walk across the spherical surface using the random walk algorithm, which simulates the ZPG realization [29]. The random walk consists of a series of forward and backward motions. The steps are segments of large circles on the spherical surface that are geodesic curves of a specific length. The turns are evenly dispersed within a specified maximum angle, plus or minus. There are only two control parameters of the model, *ω_x_* and *ω_y;_* they are the angular velocity of the *x*- and *y*-axes with tunable magnitude and direction; negative and positive sign represents clockwise and counter-clockwise axial rotation, respectively. It is found that as both magnitude and number of changing sign during Δ*t* of *ω_x_* and *ω_y_* obeys the Poisson distribution with different strengths, such a random walk converges to both uniform and nonuniform distribution over the sphere (Figure 2) to realize simulated ZPG via a large number of Monte Carlo simulations. So, the magnitude and direction of *ω_x_* and *ω_y_* are both Poisson stationary stochastic processes to generate the random walk on the spherical surface (Figure 2), satisfying *μ* = 0 and *σ* = 0 or *σ* ≠ 0 to obtain different simulated ZPG.

As shown in Figure 2, the trajectory on the upper-left subfigure is distributed evenly over the sphere, whereas the gravity vector trajectory on the other subfigures spends more or less time on specific regions of the sphere, resulting in simulated 0 gravity and partial gravity of 0.3 g, 0.5 g, and 0.7 g, respectively.

More importantly, in order to protect it from potentially harmful centrifugal acceleration, the cardiovascular system must be positioned in the middle of the experimental frames.

### 3.2. CFD Modeling of Cardiovascular System

#### 3.2.1. Geometric Configuration of Cardiovascular System

For CFD modeling and simulation, the accurate 3D geometric design of the cardiovascular system’s inner domain—which includes a portion of the aorta, its branches, and the surrounding tissue (cardiac muscle)—was developed in accordance with the anatomical properties of the system (Figure 3).

The geometric design depicts two views of the model domain, one with and one without the heart muscle, as shown in Figure 3. The heart muscle must be taken into account during the mechanical analysis because of its stiffness, which prevents arterial distortion brought on by applied pressure.

#### 3.2.2. Boundary Conditions

The same pressure boundary condition was set on cross sections of blood vessels including aorta and its branches (Figure 3), namely *p* = 125.90 mmHg. However, those pressure readings represent the averages across a heartbeat cycle. The pressure changes between a minimum and a maximum value over the course of a cycle. For the time-dependent study, the pressure distribution is changed over time using the following straightforward trigonometric function:(9)$$f\left(t\right)=\left\{\begin{array}{c} \frac{2}{3}sin\pi t,\;0\leq t\leq 0.5\;s\\ 1-\frac{2}{3}cos\left(2\pi \left(t-0.5\right)\right),\;0.5\;s\leq t\leq 1.5\;s\; \end{array}\right.$$

The first portion of the function between 0 and 0.5 s is essentially a ramp that makes it possible to determine the initial state; it has no physical importance. The second component of the function causes the pressure to fluctuate between its minimum and maximum value during the course of a heartbeat (about one second).

### 3.3. Fluid-Dynamics Analyses by CFD Simulation

According to related documents [30,31,32], the parameters in the CFD models were properly specified for CFD simulation (Table 1).

In addition, the CFD simulation options have been correctly adjusted in accordance with the problem’s complexity, accuracy, speed, and computing cost for the digital simulation (Table 2).

According to Equation (6), the shear stress (*τ*) on the vascular wall was calculated by finite element analysis as follows:(10)$$\tau =\mu \frac{\partial u_{f}}{\partial r}$$
where *r* is blood vessel diameter [33].

The grid independence test has been performed during the CFD simulation. The mesh is refined continuously until the key parameters do not change with grid quantity or their convergence precision is between 0.5% and 2%, which indicates the simulation results are independent on grid quantity (Figure 3). In order to investigate the impact of the simulated ZPG on the distribution of blood flow velocity, pressure, and shear stress on aorta walls, CFD simulations were conducted using the governing equations (Equations (3) and (6)–(8)), boundary conditions (Equation (9), and other *Dirichlet* conditions), preset parameters (Table 1), and simulation options (Table 2). The maximum values of these dependent variables in the domains of the aorta and its ramification during the CFD simulations were only illustrated in the Figure 4.

According to the CFD simulation results (Figure 4), as the simulated ZPG gradually decrease from 0.7 g to 0.5 g to 0.3 g to 0 g, the maximum values of blood flow velocity, pressure, and shear stress on the walls of the aorta and its ramification significantly increase when compared with normal gravity of 1 g.

According to Equation (10), the shear stress on aorta walls is calculated by u_f_ which is obtained from Equation (6) with term $\vec{g}\left(t\right)$ through finite element analyses, therefore shear stress is a stochastic process generated by *ω_x_* and *ω_y_* obeying the Poisson distribution (Figure 5). From Figure 5, the mean of stochastic process of shear stress on aorta walls increases significantly as simulated ZPG are gradually approaching to zero.

### 3.4. Discussions

The CFD simulations show that the blood flow velocity, pressure, and wall shear stress in the simulated ZPG appear to be higher than those in 1 g of normal gravity (Figure 5), which may indicate a cardiovascular system risk factor [34]. If there is active plaque on the walls of the aorta and its ramification, it may cause plaque detachment and acute myocardial infarction, which is highly consistent with the existing experimental phenomena and data [10,11].

To avoid or minimize health risks, we suggest that middle-aged and elderly people with active plaque in the cardiovascular system avoid being in ZPG environments, such as riding rollercoasters, parachuting, taking a parabolic flight, and so forth. Meanwhile, there also exists a cardiovascular risk for astronauts and athletes staying in ZPG in the outer-space environment, because of the higher blood velocity and pressure. Shear stress on the wall of blood vessels could cause cardiovascular disease in the long run, therefore it is necessary to take preventive measures to reduce the risk of cardiovascular disease, such as developing corresponding drugs, time, and frequency optimization for entering and staying in the ZPG environment, and so on.

## 4. Conclusions

In this article, based on simulated ZPG of 0.7 g, 0.5 g, 0.3 g, and 0 g realized through a rotating frame with a random work algorithm, a precise 3D geometric configuration of the cardiovascular system, governing equations of the *Navier-Stokes* laminar flow and solid mechanics in combination with appropriate boundary conditions, CFD simulations were conducted to investigate the influences of simulated ZPG on blood flow velocity, pressure, and shear stress in the cardiovascular system, the results show that compared with normal gravity of 1 g, under 0.7 g, 0.5 g, 0.3 g and 0 g, the blood flow velocity, respectively, increases 1.2, 1.3, 1.5 and 1.8 times in average, blood pressure is 1.1, 1.2, 1.6 and 1.7 times in average, shear stress on vascular wall is 1.3, 1.7, 2.6 and 3.2 times in average. Therefore, ZPG can greatly increase cardiovascular risk compared with normal gravity of 1 g. The research could lay a theoretical foundation for the influence of ZPG on cardiovascular risk and the development of effective prevention and control measures.

## Figures and Tables

**Figure 1 life-13-00407-f001:**
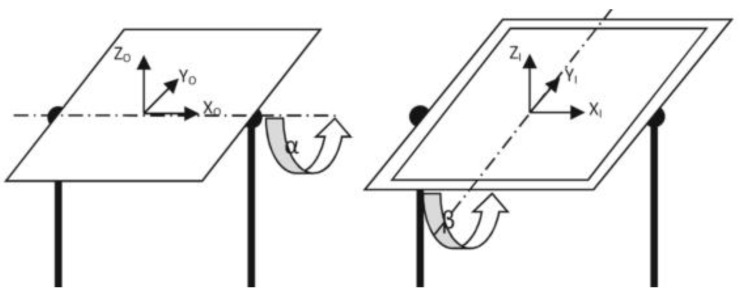
Rotating frame with two degrees of freedom (*α* and *β* denotes rotation angles of *x*- and *y*-axes, respectively).

**Figure 2 life-13-00407-f002:**
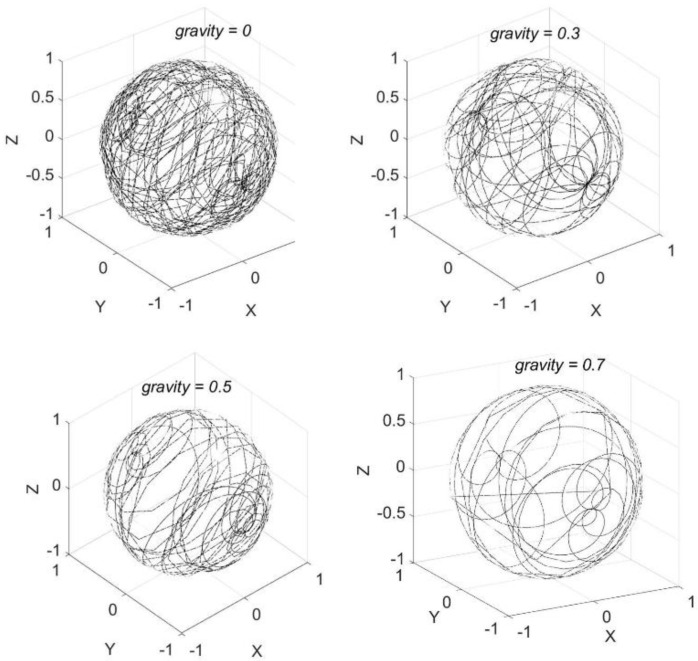
Trajectory distribution of the gravity vector over the sphere fulfilled by the axes’ rotations with angular velocities obeying the Poisson distributions for simulated ZPG.

**Figure 3 life-13-00407-f003:**
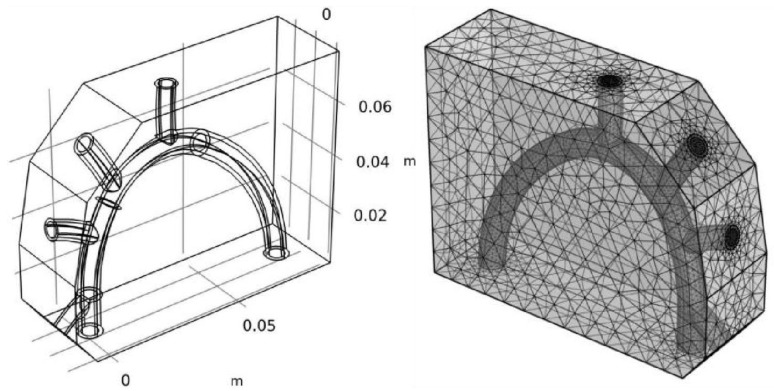
Geometric configuration of cardiovascular system and mesh generation.

**Figure 4 life-13-00407-f004:**
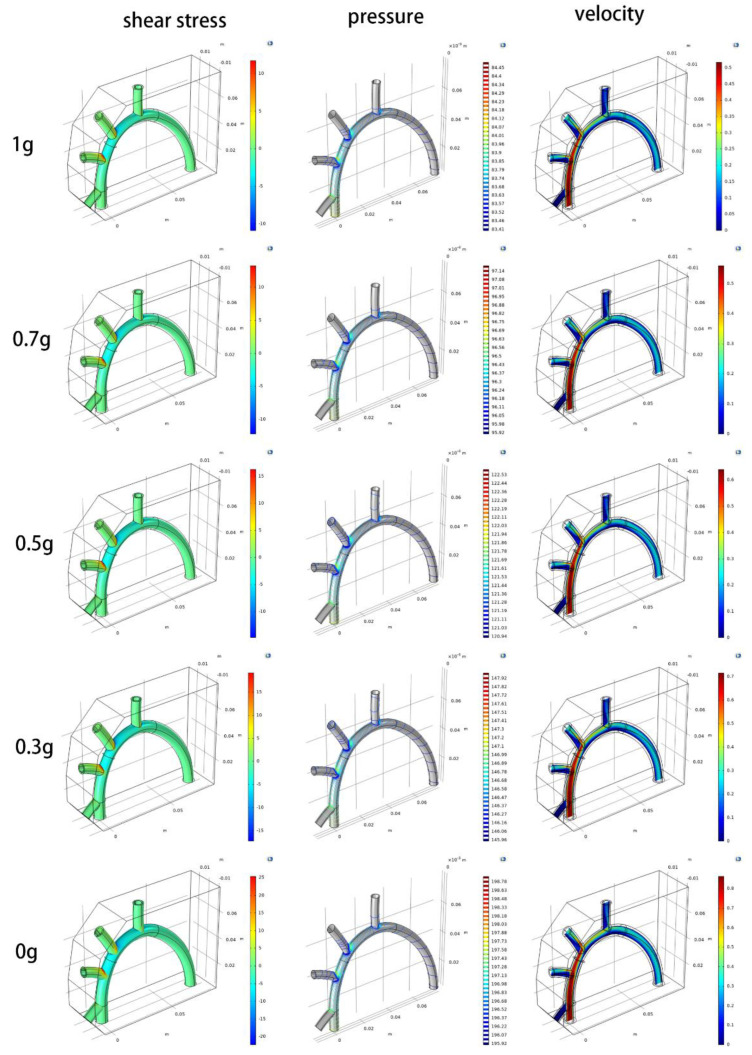
Maximum distribution value of blood flow velocity, pressure, and shear stress on aorta walls.

**Figure 5 life-13-00407-f005:**
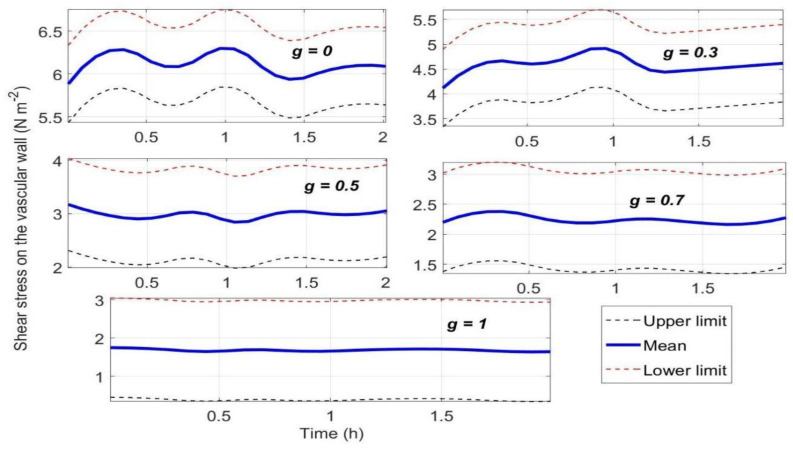
Dynamic responses of shear stress on aorta walls to gravity variations.

**Table 1 life-13-00407-t001:** Parameters specified for CFD model.

Parameter	Value	Unit	Significance
*ω_x_*	[0.25, 0.76]	Rad s^−1^	Angular velocity range of *x*-axis
*ω_y_*	[0.20, 0.78]	Rad s^−1^	Angular velocity range of *y*-axis
*a_x_*	0.26	dimensionless	Maximum number of changing in sign for *ω_x_* during 1 s
*a_y_*	0.24	dimensionless	Maximum number of changing in sign for *ω_y_* during 1 s
*T*	2	h	Terminal time of CFD simulation
*ρ*	1060	Kg m^−3^	Blood density
*μ*	5 × 10^−3^	N·s m^−2^	Dynamic viscosity of blood
*ρ_s_* _1_	960	Kg m^−3^	Artery density
*ρ_s_* _2_	1200	Kg m^−3^	Cardiac muscle density
*T*	20	℃	Temperature
*p_r_*	1	atm	Atmospheric pressure
*μ_la_*	6.2 × 10^6^	N m^−2^	Lame’ parameter of artery for linear elastic behavior
*μ_nla_*	1.24 × 10^8^	N m^−2^	Lame’ parameter of artery for nonlinear elastic behavior
*μ_lc_*	7.2 × 10^6^	N m^−2^	Lame’ parameter of cardiac muscle for linear elastic behavior
*μ_nlc_*	1.44 × 10^8^	N m^−2^	Lame’ parameter of cardiac muscle for nonlinear elastic behavior

**Table 2 life-13-00407-t002:** Configurations for digital CFD simulation.

Options	Value	Description
Number of free tetrahedral elements	10^6^	To achieve higher precision and a faster rate of convergence throughout the simulation process, relatively fewer finite elements must be used.
Element size range of entire geometry	[4.08, 8.15]	All domains’ free triangular and tetrahedral elements come in a variety of sizes.
Element size range of boundary	[3.04, 1.56]	On specific borders in Figure 3, finer grids were used to obtain a more accurate answer.
Solver	SPOOLES	For calculations and simulations, a stationary sparse object-oriented linear equation solver was used.
Relative tolerance	10^−3^	Instead of doing a set number of iterations, the solver iterates until the condition given by the relevant operation feature is satisfied.
Nonlinear method	Automatic (*Newton*)	Repeat the damping-factor reduction using an affine invariant form of the damped *Newton* technique until the relative error is lower than it was in the previous iteration.
Minimum damping factor	10^−7^	For the damped *Newton* methodology, the damping factor’s minimum value.
Maximum number of iterations	120	The most iterations that can be performed to arrive at a distinct solution.
Adaptive mesh refinement	Yes	A strategy for increasing solution precision by customizing the mesh to the physical characteristics of the issue.
Time interval of simulation	2	CFD simulation lasts 2 h

## Data Availability

Not applicable.

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
