# Peer review of "Hemodynamic Characteristics of Cardiovascular System in Simulated Zero and Partial Gravities Based on CFD Modeling and Simulation"

_life, 2023, doi:10.3390/life13020407_

Round 1
Reviewer 1 Report
The author has tried to solve an interesting problem. However, the article is lacking in many aspects. The following points need to be incorporated.
-
In the introduction section, other CFD models used earlier must be discussed for comparison where partial gravities are not taken into account.
-
Has a grid independence test been performed during simulation? Elaborate mesh quality of the model.
-
Present model can be validated with the earlier proposed model, if there is any. Justify the type of solver being used in simulation.
-
The result and discussion section needs to be explained using different contour plots at various planes.
-
The conclusion should be more quantitative.
Author Response
We would like to thank the reviewers for the valuable comments and constructive suggestions that have improved the quality and clarity of the manuscript. Revised parts are marked in the manuscript.
Reviewer 1
The author has tried to solve an interesting problem. However, the article is lacking in many aspects. The following points need to be incorporated.
- In the introduction section, other CFD models used earlier must be discussed for comparison where partial gravities are not taken into account.
Reply: Thank you very much for your positive and valuable comments. we have referenced these related researches at a suitable location in the introduction section in the revised manuscript.
- Has a grid independence test been performed during simulation? Elaborate mesh quality of the model.
Reply: Thank you very much for your reminding. We have carried out grid independence test during simulation, and given the test results in “3.3 Fluid-dynamics analyses by CFD simulation” in the revised manuscript.
- Present model can be validated with the earlier proposed model, if there is any. Justify the type of solver being used in simulation.
Reply: This suggestion is quite reasonable. Before establishing the CFD model, we referred to a large number of earlier proposed models on the theory and method of hemodynamic model, such as governing equations and boundary conditions. References to relevant literature have been included in the introduction.
According to the characteristics of the problem in this study, such as nonlinear, weakly coupled multi-physical field and time-variant gravity, SPOOLES is used. Because it has high stability, small memory consumption and high solving accuracy, the accuracy and reliability of CFD simulation results can be fully guaranteed. Although the convergence rate is a little slow, we can easily overcome this shortcoming by using our supercomputer platform.
- The result and discussion section needs to be explained using different contour plots at various planes.
Reply: Thank you for this good advice. We believe that this is a data visualization problem. In Fig.4, we have added contour line to the pressure diagram, and both blood flow velocity and shear force are closely related to pressure. Although it is easy to add contour lines to shear stress and velocity graphs in Comsol Multiphysics, However, we found that after contour plots were added to the shear stress and velocity plots in Fig.4, two colorbars must be used to represent the spatial distribution of shear stress and velocity, and their display effect is not as clear as that only with heat map. Therefore, in order to have a clearer display effect, we only added contour line to pressure, while heat maps were still used for shear stress and velocity graphs.
- The conclusion should be more quantitative.
Reply: Thank you very much for your positive and valuable comments. Although quantitative Results are given in 3. Results and discussion, we have added some additional quantitative results in the conclusion to make it more quantitative.

Reviewer 2 Report
The authors present an interesting topic aimed at modelling the effect of different gravitational effects on cardiovascular risk. This paper will be of interest to the reader. However, I am more concerned about several aspects of this paper.
Main issues.
Introduction.
1. the authors' study focuses on the cardiovascular effects of different gravitational effects, then it should focus on the simulation studies of different gravity states and the related physiological effects on humans including cardiovascular, and the principles of how different tilt angles simulate different gravitational effects should be elaborated. It should also be presented in the context of previous studies on the effects of different gravities on the cardiovascular system.
2. Another focus of the study is CFD modelling and the authors might have described and presented in detail the relevant studies on CFD modelling to better introduce their research topic.
3. The authors indicate in the last paragraph that CFD models do not currently take into account the effects of spatial gravity. However, the specific research hypothesis and the issues to be addressed by the manuscript are not clearly stated.
Materials and methods.
1. The authors spell out the way and method of the different gravities simulated in the simulated part of gravity. And it does not specify what gravity was simulated, what angle of inclination was used for the simulation, and for how long. Typically, the subject of gravitational effects on the cardiovascular system is the human, so several control quantities need to be considered for different human-centred gravity simulations, including what gravity is simulated? How is it simulated? Only by controlling these basic variables can we simulate different gravities and the associated elements accordingly.
Current studies of simulated gravity effects typically simulate microgravity in orbit, low gravity on the Moon or Mars and normal Earth gravity, which is more widely studied in current aerospace medicine and medical engineering, and it is suggested that the authors could analyse articles in this category to more adequately set the simulation conditions.
2. The authors describe the modelling of gravity and the modelling and simulation of the cardiovascular system in their methods, but do not specify what was used to perform the simulations? Equipment and instrumentation also need to be clarified in detail, and showing relevant pictures is necessary.
3. CFD modelling also needs to be highlighted in the methods.
Results and discussion
When reporting results and discussion, authors should take care to report specific results first, especially the highlights of the results, which contain the key findings of the study. This is followed by the need to validate previous studies against the results. The author's presentation in this section is confusing and I strongly recommend that the author expresses the results and discussion separately rather than mixing them together.
General comments
1. The author's title expresses the discovery or study of a method and uses simulations throughout the text for validation. However, it cannot be appropriate to define a method through one or several simulation validations. The proposal and definition of a completely new method requires real-life experiments in addition to computer simulations and modelling analysis, and in the field of aerospace research, at least ground-based simulation experiments are required to verify the reliability and validity of the technology. I would therefore suggest that the authors carefully consider the direction in which the article should be written.
2. The author should seriously consider the idea of writing the manuscript. From the current logic of reading the manuscript, the manuscript seems to be a kind of computational report, and the author should seriously consider elaborating and reporting in detail the purpose, method, discussion of findings, and conclusion focus so that the reader can understand the details of the author's experiments more clearly.
3. The author should revise the English language and style of the paper in depth to reduce the presence of colloquialisms and unnecessary modifiers. For example, in the abstract and results, the phrase "a large number of computational fluid dynamics simulations were carried out" is often used, which should not appear in a scientific manuscript, and the magnitude of the calculations should be expressed in the methods and results in terms of objective data and content rather than subjective interpretation.
The authors are advised to revise the manuscript in depth to improve the quality of the manuscript.
Author Response
Detailed Response to Reviewer Comments
We would like to thank the reviewers for the valuable comments and constructive suggestions that have improved the quality and clarity of the manuscript. Revised parts are marked in the manuscript.
Reviewer 2
The authors present an interesting topic aimed at modelling the effect of different gravitational effects on cardiovascular risk. This paper will be of interest to the reader. However, I am more concerned about several aspects of this paper.
Main issues.
Introduction.
- the authors' study focuses on the cardiovascular effects of different gravitational effects, then it should focus on the simulation studies of different gravity states and the related physiological effects on humans including cardiovascular, and the principles of how different tilt angles simulate different gravitational effects should be elaborated. It should also be presented in the context of previous studies on the effects of different gravities on the cardiovascular system.
Reply: This suggestion is quite reasonable. In introduction, we have referenced previous studies of different gravity states and the related physiological effects on humans as well as how to simulate microgravity effect on the ground in “3.1 Simulated ZPG realized by random walk” in the revised manuscript.
- Another focus of the study is CFD modelling and the authors might have described and presented in detail the relevant studies on CFD modelling to better introduce their research topic.
Reply: Thank you for your comments. We have referenced relevant studies on CFD modelling and rewritten the corresponding part in introduction to highlight the gap between our work and other studies, and show our contribution in the revised manuscript.
- The authors indicate in the last paragraph that CFD models do not currently take into account the effects of spatial gravity. However, the specific research hypothesis and the issues to be addressed by the manuscript are not clearly stated.
Reply: Thank you for this good advice. We have elaborate on research hypothesis and important issues that needs to be addressed in the revised manuscript, according to your comments.
Materials and methods.
- The authors spell out the way and method of the different gravities simulated in the simulated part of gravity. And it does not specify what gravity was simulated, what angle of inclination was used for the simulation, and for how long. Typically, the subject of gravitational effects on the cardiovascular system is the human, so several control quantities need to be considered for different human-centred gravity simulations, including what gravity is simulated? How is it simulated? Only by controlling these basic variables can we simulate different gravities and the associated elements accordingly.
Current studies of simulated gravity effects typically simulate microgravity in orbit, low gravity on the Moon or Mars and normal Earth gravity, which is more widely studied in current aerospace medicine and medical engineering, and it is suggested that the authors could analyse articles in this category to more adequately set the simulation conditions.
Reply: Thank you for this good advice. There are only two control parameters of the model, ωx and ωy that is angular velocity of x- and y-axis, respectively, whose magnitudes and directions are variable —— negative sign for clockwise rotation and positive sign for counter-clockwise rotation. Through theoretical study, It is found that as both magnitude and number of changing sign during â–³t of ωx and ωy obeys the Poisson distribution with different strengths, such a random walk converges to both uniform and nonuniform distribution over the sphere (Fig.2) to realize simulated zero and partial gravities (ZPG) via a large number of Monte Carlo simulations. so the magnitude and direction of ωx and ωy is both Poisson stationary stochastic processes to generate the random walk on the spherical surface (Fig.2), satisfying μ = 0 and σ = 0 or σ≠0 to obtain different simulated ZPG.
As the research object is human, we refer to the relevant literature on the study of astronauts in orbit for the determination of ωx and ωy. In the revised draft, we gave magnitude range, maximum number of changing in sign during 1 second for ωx and ωy, as well as simulation time in the Table 1. Through CFD simulation, the results show that this range can make the blood pressure, blood flow and shear force of the vessel wall fall within a reasonable range.
- The authors describe the modelling of gravity and the modelling and simulation of the cardiovascular system in their methods, but do not specify what was used to perform the simulations? Equipment and instrumentation also need to be clarified in detail, and showing relevant pictures is necessary.
Reply: This suggestion is quite reasonable. It's worth mentioning that we didn't carried out prototype experiments, so we didn't have instrumentation pictures. We verify our assumptions just based on numerical simulation, and the results are used to explain some phenomena, For example, some older people suffered sudden myocardial and cerebral infarction during they rode roller coaster, pendulum ride and helter-skelter in amusement park, etc. Although our research ultimately requires prototype experiments for final confirmation, in view of the complexity of prototype experiments and the ethical issues involved, we do not have the conditions to do prototype experiments at present, so we can only do computer simulation to carry out theoretical research on this issue.
Since the NS governing equation of blood flow, the structural equation of myocardial tissue and the simulation of microgravity effect by rotating device are all reliable theories and methods, we only coupled them closely to study the changes of dynamic characteristics of blood flow under different simulated microgravity effects, so it can be considered that the simulation research results are reliable and effective. It can provide scientific and beneficial reference for understanding cardiovascular health problems of human in ZPG.
- CFD modelling also needs to be highlighted in the methods.
Reply: Thank you very much for your reminding. We have added a description of CFD modeling in the methods, but a more detailed modeling process is placed in the results section.
Results and discussion
When reporting results and discussion, authors should take care to report specific results first, especially the highlights of the results, which contain the key findings of the study. This is followed by the need to validate previous studies against the results. The author's presentation in this section is confusing and I strongly recommend that the author expresses the results and discussion separately rather than mixing them together.
Reply: Thank you very much for your positive and valuable comments. We have separated the discussion section separately to form a new 3.4 discussion section, and provided specific results, validate previous studies against the results, etc.
General comments
- The author's title expresses the discovery or study of a method and uses simulations throughout the text for validation. However, it cannot be appropriate to define a method through one or several simulation validations. The proposal and definition of a completely new method requires real-life experiments in addition to computer simulations and modelling analysis, and in the field of aerospace research, at least ground-based simulation experiments are required to verify the reliability and validity of the technology. I would therefore suggest that the authors carefully consider the direction in which the article should be written.
Reply: Thank you very much for your reminding. We carefully revised the paper to emphasize the use of computer experiments to verify scientific hypothesis, rather than based on prototype experiments. Such a research method is scientific and reasonable, for reasons already described in detail, see the answer to the second question in Materials and methods part.
- The author should seriously consider the idea of writing the manuscript. From the current logic of reading the manuscript, the manuscript seems to be a kind of computational report, and the author should seriously consider elaborating and reporting in detail the purpose, method, discussion of findings, and conclusion focus so that the reader can understand the details of the author's experiments more clearly.
Reply: Thank you for your comments. For readers to understand our works more clearly, we add the hypothesis in the introduction. The whole paper is mainly organized in five parts: hypothesis, CFD modeling, experiment design of CFD simulation, CFD simulation for verification of hypothesis, discussion and conclusion. Meanwhile, we elaborated and reported in detail the purpose, method, discussion of findings, and conclusion focus in the revised manuscript.
- The author should revise the English language and style of the paper in depth to reduce the presence of colloquialisms and unnecessary modifiers. For example, in the abstract and results, the phrase "a large number of computational fluid dynamics simulations were carried out" is often used, which should not appear in a scientific manuscript, and the magnitude of the calculations should be expressed in the methods and results in terms of objective data and content rather than subjective interpretation.
Reply: Thank you for your comments. We have substantially revised the writing of the paper and removed the expression "a large number". To be frank, in the process of simulating partial gravities, a large number of computer simulations are indeed carried out. For example, a large number of Monte Carlo simulations were performed to get ZPG of 0.7g, 0.5g, 0.3g and 0g.
The authors are advised to revise the manuscript in depth to improve the quality of the manuscript.
Reply: Thank you very much for your positive and valuable comments, we have substantially revised the manuscript.
Thanks to all reviewers for the thoughtful and thorough review. Hopefully we have addressed all of your concerns.

Round 2
Reviewer 1 Report
The article can be accepted for publication.